# Drug-Free Nasal Spray as a Barrier against SARS-CoV-2 and Its Delta Variant: In Vitro Study of Safety and Efficacy in Human Nasal Airway Epithelia

**DOI:** 10.3390/ijms23074062

**Published:** 2022-04-06

**Authors:** Fabio Fais, Reda Juskeviciene, Veronica Francardo, Stéphanie Mateos, Manuela Guyard, Cécile Viollet, Samuel Constant, Massimo Borelli, Ilja P. Hohenfeld

**Affiliations:** 1Altamira Medica AG, 6300 Zug, Switzerland; faf@altamiratherapeutics.com (F.F.); rj@aurismedical.com (R.J.); vef@aurismedical.com (V.F.); 2Texcell SA, 91000 Evry, France; mateos@texcell.fr (S.M.); manuela@texcell.fr (M.G.); cviollet@texcell.fr (C.V.); 3Epithelix Sarl, 1228 Geneva, Switzerland; samuel.constant@epithelix.com; 4Life Sciences and Technologies Department, School of PhD Programmes, Magna Graecia University, 88100 Catanzaro, Italy; massimo.borelli@gmail.com

**Keywords:** SARS-CoV-2, nasal spray, prophylactic, treatment, drug-free, medical device, protection, barrier, nasal epithelium, bentonite

## Abstract

The nasal epithelium is a key portal for infection by respiratory viruses such as SARS-CoV-2 and represents an important target for prophylactic and therapeutic interventions. In the present study, we test the safety and efficacy of a newly developed nasal spray (AM-301, marketed as Bentrio) against infection by SARS-CoV-2 and its Delta variant on an in vitro 3D-model of the primary human nasal airway epithelium. Safety was assessed in assays for tight junction integrity, cytotoxicity and cilia beating frequency. Efficacy against SARS-CoV-2 infection was evaluated in pre-viral load and post-viral load application on airway epithelium. No toxic effects of AM-301 on the nasal epithelium were found. Prophylactic treatment with AM-301 significantly reduced viral titer vs. controls over 4 days, reaching a maximum reduction of 99% in case of infection from the wild-type SARS-CoV-2 variant and more than 83% in case of the Delta variant. When AM-301 administration was started 24 h after infection, viral titer was reduced by about 12-folds and 3-folds on Day 4. The results suggest that AM-301 is safe and significantly decelerates SARS-CoV-2 replication in cell culture inhibition assays of prophylaxis (pre-viral load application) and mitigation (post-viral load application). Its physical (non-pharmaceutical) mechanism of action, safety and efficacy warrant additional investigations both in vitro and in vivo for safety and efficacy against a broad spectrum of airborne viruses and allergens.

## 1. Introduction

The COVID-19 pandemic has had a massive toll on daily life [1] and has strained the capacity of healthcare institutions [2,3,4]. The pandemic continues to have a major impact on daily conduct in our attempt to prevent the spread of the causative agent severe acute respiratory syndrome coronavirus 2 (SARS-CoV-2). This virus is the third coronavirus known to cause severe disease in humans, and its emergence follows two global outbreaks by SARS in 2002–2003 and the Middle East respiratory syndrome-related coronavirus in 2012 (reviewed in [5]). To resolve the current pandemic, governments worldwide have variably implemented a series of non-pharmaceutical interventions, from physical distancing, school closures and travel restrictions to better hygiene and face mask requirements [6,7]. Despite ongoing vaccination campaigns, there is a pressing need for new, effective personal protection measures against the infection. Moreover, the emergence and spread of SARS-CoV-2 variants is a major threat to public health. These “Variants of Concern” (VOC) have the potential to evade immunity following vaccination or recovery from infection [8], and breakthrough cases have been emerging at an increasing rate [9]. Of the variants, the most concerning in terms of severity of illness is the Delta variant [10].

SARS-CoV-2 infection is mainly contracted from airborne virions [11,12]. Upon inhalation, the virions bind to cells that express angiotensin-converting enzyme 2 (ACE2), and after host-cell proteases cleave the viral spike protein, the virions enter cells and cause infection [13]. Therefore, the main site of initial infection is the nasopharyngeal epithelium and, in particular, the ciliated cells, which express high levels of ACE2, and proteases TMPRSS2 and furin on their apical side [14]. The nose is not only a portal for viral infection but also an important site for viral replication [15]. Indeed, several studies have described a high viral load in the nasal tract during the first days of SARS-CoV-2 infection [14,16,17]. Importantly, since nasal swabs from both symptomatic and asymptomatic patients have been found to contain high viral loads [18,19,20], the nose can be considered an important target for both prophylactic and therapeutic interventions against SARS-CoV-2 infection.

An interesting approach to develop simple and safe interventions of this type is based on enhancing the protective function that the nasal barrier already exerts. Indeed, the airway epithelium of the nasal mucosa works as a physical barrier through the production of mucus, which traps pathogens. Thereafter, the clearing action of cilia discharges the mucus into the nasopharynx from where it is eventually swallowed [21]. A second line of protection is provided by immune cells resident in the nasopharynx-associated lymphoid tissue [15]. Together, mucociliary clearance and immune responses should protect the nasal epithelium from pathogens, but infection can ensue in cases of high viral exposure or dysfunction of these mucosal defenses.

Several nasal sprays are commercially available for the treatment of respiratory infections such as common cold and influenza (see, for example, Carragelose^®^-based nasal spray, Nasaleze^®^ Cold & Flu blocker, Vicks^®^ First Defence Cold Virus Blocker), and recent research has investigated the possibility of repurposing these sprays as COVID-19 prophylactic agents. A previous study [22] observed that two products containing carrageenan, but no pharmaceutically active ingredients, inhibited SARS-CoV-2 infection in in vitro models of air–liquid interface (ALI) cultures of human airway epithelial cells (HAECs) and human lung cells (Calu-3). This same study found that sprays with other inert or active ingredients were cytotoxic. Carrageenans are polyanionic, sulfated polysaccharides from red seaweed; they are widely used in pharmaceutical formulations and have known virus-binding properties [23]. Other natural substances with broad pharmaceutical applications and virus binding properties are clays, including bentonite [24,25,26,27]. Bentonite is a clay mineral composed of thin aluminum silicate sheets with a high specific surface with a net negative charge; these properties contribute to its ability to bind viral particles and molecules such as drugs [26]. We therefore hypothesized that a bentonite-containing nasal spray could protect against SARS-CoV-2 and other airborne pathogens. Bentonite suspensions can have thixotropic properties, that is, they reversibly change from a gel when undisturbed to a fluid colloid when agitated [28]. We envisioned a bentonite-containing nasal spray that could be applied as a liquid, but in the nasal cavity, it would form a durable, protective gel barrier.

We therefore devised a nasal spray formulation (AM-301) with the aim of providing a safe and effective means of protection against harmful airborne particles. Seeking to minimize potential side effects and to facilitate frequent and compliant use, AM-301 is composed of only inert ingredients such as pharmaceutical excipients and substances that are generally recognized as safe. Because the components are neither metabolized nor absorbed and the mechanism of action is physical, not chemical, AM-301 is a non-pharmaceutical medical device from a regulatory perspective. This study tests the ability of AM-301 to prevent SARS-CoV-2 infection or mitigate existing infections in vitro using a model of primary human nasal airway epithelium (MucilAir, Epithelix, Geneva, Switzerland).

## 2. Results

### 2.1. In Vitro Safety

Inserts of a 3D model of the primary human nasal airway epithelium (MucilAir) were repeatedly exposed to AM-301 or its matrix (lacking bentonite), or left untreated for 4 days, and then tested in four standard assays to evaluate MucilAir integrity. TEER measurements (Figure 1A) at baseline were in the normal range (200–600 Ω cm^2^) for all inserts, indicating their suitability for use in toxicology testing. Mean values for inserts treated with AM-301 or the matrix and for untreated inserts increased slightly over 4 days. These results indicate that the product did not have any adverse effects on tissue integrity (2-way repeated measures ANOVA: treatment, F_(2,6)_ = 29.30, *p* = 0.0008; time, F_(2.112, 12.67)_ = 153.0, *p* < 0.0001; interaction F_(8,24)_ = 15.76, *p* < 0.0001. Post hoc Tukey’s test: Day −3: untreated vs. matrix, *p* > 0.05; untreated vs. AM-301, *p* > 0.05; matrix vs. AM-301, *p* > 0.05; Day 1: untreated vs. matrix, *p* > 0.05; untreated vs. AM-301, *p* = 0.0143; matrix vs. AM-301, *p* = 0.0264; Day 2: untreated vs. matrix, *p* = 0.0057, untreated vs. AM-301, *p* > 0.05, matrix vs. AM-301, *p* = 0.0005; Day 3: untreated vs. matrix, *p* > 0.05, untreated vs. AM-301, *p* = 0.0049, matrix vs. AM-301, *p* = 0.0210; Day 4, untreated vs. matrix, *p* = 0.0476; untreated vs. AM-301, *p* = 0.0082, matrix vs. AM-301, *p* > 0.05).

The release of LDH to the basolateral medium, a sign of cell lysis, was assayed after 2 and 4 days of exposure to the products (Figure 1B). At both time points, the normalized percentage of cytotoxicity was well below 5% for treated and untreated inserts alike (unpaired *t*-test, matrix vs. AM-301: Day 2 (*p* = 0.776); Day 4 (*p* = 0.392). These results indicate that the product had no acute cytotoxic effects on cells and that, in all treatment conditions, only normal cell turnover occurred (values ≤ 5%). After 4 days, cilia beating frequency at the apical surface was measured (Figure 1C). Untreated inserts had a mean frequency of 4.6 Hz, while inserts treated with the matrix or AM-301 had slightly lower values (3.5 and 3.7 Hz; unpaired *t*-test, *p* = 0.011 and *p* = 0.025, respectively).

### 2.2. In Vitro Efficacy

#### 2.2.1. Pre-Viral Load (Prophylactic) Efficacy

To test AM-301′s ability to protect against SARS-CoV-2 infection of the nasal epithelium, MucilAir inserts were treated apically with the product, its matrix, or physiological saline shortly before exposure to the virus (wild-type or Delta variant). The application of the product was repeated daily for 4 days (Figure 2A,B). In case of infection by the wild-type (WT) variant, viral replication was robust in both saline- and matrix-treated inserts, with more than 10-fold daily increases in titer from Days 1 to 3 and a smaller increase on Day 4. In contrast, in AM-301-treated inserts, viral replication was strongly dampened. On Day 4, a 2-log reduction in TCID_50_ compared with saline control was observed, which corresponds to about a 99% lower viral titer (Figure 2A). 

AM-301 was shown to be effective also when inserts were infected with the SARS-CoV-2 Delta variant, with a six-fold reduction in viral titer compared with saline and matrix-treated groups on Day 3, and of six- and three-folds on Day 4, respectively (Figure 2B).

Viral titer for the inserts not exposed to SARS-CoV-2 was below detection, indicating that the test substances and culture medium were free of viral contamination (data not shown for the sake of clearness in the graphical representation).

Because viral replication was unhindered in the matrix-treated samples, we can infer that bentonite within the AM-301 formulation is primarily responsible for the effect.

The data from the above prophylaxis experiments (pre-viral load application of AM-301) were statistically analyzed using a linear mixed-effect model (Figure 2C,D). The time profile of SARS-CoV-2 infection in inserts that received a prophylactic treatment with AM-301 was significantly decelerated compared with that of inserts that received saline or matrix, both when the infection was caused by the WT and by the Delta variants (Figure 2C,D, respectively). In case of the WT variant, the saline and matrix experimental conditions showed significantly faster viral titer growth compared with the one observed in AM-301-treated inserts (t = 5.13; *p* < 0.001). AM-301 was able to significantly decelerate viral titer growth also in case of infection by the Delta variant, compared with inserts receiving saline solution or the matrix (Figure 2D, t = 4.69, *p* < 0.001).

#### 2.2.2. Post-Viral Load (Mitigation) Efficacy

To test AM-301’s ability to mitigate an existing SARS-CoV-2 infection of the nasal epithelium, MucilAir inserts were infected and then treated with test substances starting 24 h after infection (Figure 3A,B). A high viral titer was observed in saline- and matrix-treated samples, while the AM-301-treated inserts showed a lower viral titer, both in case of infection by SARS-CoV-2 WT and by the Delta variant (Figure 3A,B). In case of infection by SARS-CoV-2 WT, at the end of the treatment period (Day 4), inserts that had received AM-301 showed significantly lower viral titer (12- or 8-fold lower, respectively) than saline- or matrix-treated inserts (Figure 3A).

AM-301 was effective at reducing the viral titer also in the case of infection by the SARS-CoV-2 Delta variant (Figure 3B), reaching a maximum effect on Days 3 and 4. Indeed, on Day 3, viral titer measured in inserts treated with AM-301 was 6.6-fold lower and 7.3-fold lower than the one measured in inserts receiving saline solution or matrix, respectively. On Day 4, a more moderate reduction was observed, reaching 3.1- and 3.3-fold reductions compared with the saline- and matrix-treated groups, respectively.

The viral titer for the inserts not exposed to SARS-CoV-2 was below detection, indicating that the test substances and culture medium were free of viral contamination (data not shown).

As in the prophylaxis experiment (pre-viral load application), data from the mitigation arm of the study (post-viral load application) were also analyzed with a linear mixed-effect model (Figure 3C,D), which revealed that (i) the time profile (replication kinetics) of SARS-CoV-2 titer was not impacted by matrix or saline solution; (ii) the time profile of SARS-CoV-2 titer in the presence of AM-301 was significantly decelerated (SARS-CoV-2 WT, t-value 2.50, *p* < 0.05; Delta variant, t-value 4.08, *p* < 0.001) compared with inserts that received saline solution or matrix. Importantly, all experimental conditions were statistically equivalent at baseline. All data were considered valid, and no outliers were excluded.

Altogether, despite noticeable intragroup variability, these results suggest that AM-301 can mitigate an established infection even when applied several hours post-exposure to the virus, both in the cases of the SARS-CoV-2 WT and the Delta variant.

## 3. Discussion

To study the ability of AM-301 to prevent or reduce SARS-CoV-2 infection in the nasal mucosa, we used a well-established model of primary human nasal epithelium also for viral infections, including SARS-CoV-2: MucilAir [29,30,31,32]. AM-301 was studied to determine its safety in MucilAir, its efficacy in preventing MucilAir from being infected by SARS-CoV-2, and its ability to mitigate an established infection in MucilAir without any previous treatment.

In the first part of the study, we performed in vitro safety assays (TEER, LDH, and CBF) to assess potential toxicity issues of AM-301. The results displayed in Figure 1A–C show that AM-301 and its matrix (lacking bentonite) had no toxic effects on MucilAir inserts despite repeated application over 4 days: the measures of tight junction integrity in treated cultures did not differ from those of untreated cultures, and an LDH assay of cytotoxicity revealed no increase in cell death. A slight reduction in ciliary beating frequency (CBF) was detected in AM-301 and the matrix-treated inserts compared with the controls (Figure 1C). However, there was no difference between AM-301 and its matrix, suggesting that bentonite is not responsible for this effect, which may rather be due to the viscosity of the formulation. Importantly, safety and tolerability of AM-301 were further confirmed in a clinical investigation with 36 allergic rhinitis patients, where AM-301 was well tolerated and was considered safe for human use [33].

In the second part of the study, we focused on the evaluation of AM-301 as a potential prophylaxis or treatment against SARS-CoV-2 infection. To investigate that, we used the same in vitro model of human nasal airway epithelium (MucilAir), receiving AM-301 application either pre-viral load (Figure 2, to simulate a prophylactic use) or post-viral load (Figure 3, to simulate a mitigation use post-exposure). The efficacy of AM-301 was tested both against the SARS-CoV-2 WT and the SARS-CoV-2 Delta variant. Because of technical reasons (i.e., availability of the SARS-CoV-2 Delta variant), the infection with SARS-CoV-2 Delta variant occurred at MOI 0.1, 5-fold lower than the one used for the SARS-CoV-2 WT (MOI = 0.5). This detail must be taken into account when comparing the viral load throughout the experiment (Figure 2A,C vs. Figure 2B,D and Figure 3A,C vs. Figure 3B,D).

The results shown in Figure 2 indicate that pre-viral load application of AM-301 (but not of its matrix) on MucilAir inserts was protective against SARS-CoV-2 WT infection, as just a daily application of the product led to a 2-log (99%) reduction in viral titer by Day 4 (Figure 2A). AM-301 was shown to be able to also protect against infection by the SARS-CoV-2 Delta variant. Indeed, the viral titer measured in inserts receiving AM-301 was six-fold lower than the one for saline- or matrix-treated inserts already on Day 3, and six- and three-folds lower (respectively) at the end of the experiment (Figure 2B). In order to specifically evaluate the effect of AM-301 on the kinetics of the viral titer growth, we used the linear mixed-effects model [34,35], which showed a significant deceleration in viral titer growth for both the SARS-CoV-2 WT and Delta variants (Figure 2C,D, respectively). Importantly, AM-301 also showed efficacy in reducing viral titer load (Figure 3A,B) and in decelerating viral titer growth (Figure 3C,D) when the infection had already occurred (Figure 3, post-viral load application). Indeed, on Day 4, inserts that received AM-301 application 24 h post-infection showed a 12- or 8-fold reduction in the viral titer compared with the saline- and matrix-treated inserts, respectively, when the infection was elicited by the SARS-CoV-2 WT and and 3.1 and 3.3-fold reductions, respectively, when the infection was elicited by the SARS-CoV-2 Delta variant (Figure 3A,B).

The effects exerted by AM-301 can be attributed to a mechanical, not biological, action. Indeed, AM-301 was developed to mechanically prevent the virus from contacting the nasal mucosa and infecting the upper respiratory airways and to trap or bind it for clearance by mucociliary clearance, ultimately helping to prevent a dramatic viral replication and spread in the airways. This proposed mechanism is based on the hypothesis that viral particles are bound by the complex gel mesh of AM-301 stabilized by bentonite particles and not exclusively due to the presence of polyanionic substances [36,37,38]. The advantage of this type of non-pharmacological principle is that, potentially, AM-301 may have a broad spectrum of action on viruses and allergens, which can be trapped as soon as they enter the body by passing through the nose.

AM-301 is a substance-based medical device that contains only excipients and inert ingredients generally recognized as safe. Clays such as bentonite, a key component of the formulation, can also be used to detoxify water of fluoride or heavy metals [26], and bentonite is the basis of an oral treatment for acute infectious diarrhea (diosmectite, Smecta [39]). Its virus-binding properties have been known for many years [25,40,41,42], but to our knowledge, this is its first application used to protect against airborne viruses. Concerns on the possibility of AM-301 reaching the lungs are substantiated by reports of lung toxicity induced by clays such as bentonite [43,44]. AM-301 application via a nasal spray was therefore designed to ensure that the deposition of AM-301 is exclusively local, in the nasal cavity. The spray’s droplet size distribution was tuned to be well above the threshold of 10 μm to minimize particle deposition in the lungs [45]. AM-301 formulation characterization, nasal deposition pattern, and nasal residence time data are the objectives of a separate study (manuscript in preparation).

Bentonite in synergy with other formulation components confers thixotropic properties to AM-301, permitting its easy application with a nasal spray pump, which results in a protective film once it contacts the nasal epithelium. The lack of preservatives, decongestants, and other pharmacologically active molecules was intended to ensure maximal safety and, potentially, compatibility with the nasal microbiome. Further investigations to confirm this would be of particular interest. Indeed, this feature may be particularly important, since a reduction or imbalance, most recently observed also for SARS-CoV-2, in nasal microbiota diversity provoked by influenza seem to be associated with pathological conditions of the respiratory tract, such as chronic rhinosinusitis, asthma, bronchiolitis, allergic rhinitis, and otitis media [46,47]. Therefore, these features of AM-301 suggest that its use is unlikely to alter the nasal microbiome, maintaining the physiological integrity of the nasal epithelial barrier [48].

Conducting this study was challenged by several factors that may account for the noticeable intragroup variability: differences in the mucus quantities produced by the MucilAir inserts and technical difficulties in the washing procedures to evenly remove the mucus from all inserts. However, these in vitro results highlight the potential and relevance of AM-301, since the MucilAir tissue model can be considered a worst-case scenario in terms of protection from a viral infection compared with the in vivo situation. Indeed, MucilAir lacks a supportive immune system to protect against infection, and mucociliary clearance of viral particles does not occur. AM-301 was applied once every 24 h, whereas in patients, 2–3 administrations per day would be likely. Indeed, clinical data from an allergen exposure chamber study showed that AM-301 well tolerated and provided a protective effect lasting more than 3 h [33]. The encouraging results of these studies in which the effect of AM-301 was shown for both viruses and allergens imply a mechanism of action likely to be applicable to diverse airborne particles and call for further investigations in vivo and in humans, to further evaluate AM-301 as a medical device with a broad spectrum of action against as other viruses, allergens, and possibly pollutants.

Intriguingly, since nasal sprays such as AM-301 are not absorbed by the nasal mucosa and are discharged from the nose to the pharynx (manuscript in preparation), AM-301 may also exert its beneficial virus blocking effects in the throat. Since the oral cavity is the main production site of aerosols and airborne droplets [49,50], this action might decrease the potential for viral transmission to other people by talking or coughing [51]. The effect of AM-301 on viral load is currently under clinical investigation.

Currently, the entire world must deal with the health, social and economic damage that the COVID-19 pandemic caused. Several therapies and vaccines have been developed, allowing us to gradually return to normal life. The ability of SARS-CoV-2 to mutate quickly and the high cost of medications, vaccines, and protective measures as well as supply chain challenges make this process even more difficult in developing countries [52,53]. It is thus critical to have efficacious strategies that can contrast viral spread rapidly and that are helpful in contrasting different viral strains. Furthermore, its stability at high temperature, tested and confirmed in accordance with relevant guidelines (manuscript in preparation), allows for its use in warm climates. This preliminary study suggests that AM-301 could be a safe, non-pharmacological, easy-to-use nasal spray that could reduce the risk of infection from SARS-CoV-2 and potentially from other airborne viruses by acting as an “intranasal mask”. It appears to be a promising strategy for self-protection against airborne viruses and as a complement to existing preventive measures such as increased hygiene, physical distancing, and vaccination.

## 4. Materials and Methods

### 4.1. Virus and Cell Cultures

The SARS-CoV-2 strain 2019-nCOV/Italy-INMI1 was obtained from the National Institute for Infectious Diseases Lazzaro Spallanzani IRCCS (Rome, Italy) [54]. The SARS-CoV-2 strain hCoV-19/USA/PHC658/2021 (Lineage B.1.617.2; Delta Variant) was obtained from Bei Resources. The virus was propagated on VERO cells, collected, aliquoted, and stored at −70 °C until use at Texcell (Evry, France). For viral titration assays, VERO cells were cultured in DMEM supplemented with 4% fetal bovine serum (FBS). The cell line had been obtained by Texcell from the Pasteur Institute (Paris, France). Testing for mycoplasma contamination using the MycoTOOL Mycoplasma Real-Time PCR Kit (Roche Diagnostics GmbH, Mannheim, Germany) was negative.

MucilAir Pool tissue cultures were obtained from Epithelix (Geneva, Switzerland) [54,55,56]. MucilAir Pool consists of human airway epithelial cells collected from 14 healthy adult donors (male and female) and reconstituted as a 3D tissue in a two-chamber system; for this study, only nasal epithelial cells were used. The cells were cultured at 37 °C (humidified 5% CO_2_ atmosphere) on Costar Transwell porous inserts (0.33 cm^2^ each, 24-well plates) with MucilAir serum-free culture medium (cat. no. EP04MM, Epithelix). Approximately one month after seeding, when the cultures were fully differentiated and pseudo-stratified, with basal cells, ciliated cells, and mucus cells, they were exposed to air on the apical surface and were considered suitable for experimental use. Prior to testing, MucilAir Pool cultures (hereafter “MucilAir inserts”) were cultured with 0.7 mL culture medium in the basolateral chamber and air at the apical surface; the medium was changed every 3 days. Each insert had approximately 500,000 cells.

### 4.2. Nasal Spray Formulation

The nasal spray formulation tested (AM-301) in this study is a medical device containing bentonite (magnesium aluminum silicate) in a matrix composed of mono-, di- and triglycerides; propylene glycol; xanthan gum; mannitol; disodium EDTA; citric; acid and water. All components are listed in the Inactive Ingredient Database of the US Food and Drug Administration (FDA), are classified as “generally recognized as safe”, or are approved for use as food additives by the FDA. The formulation is a white to light beige, aqueous gel emulsion with a pH of 6.0. AM-301 is odorless and tasteless. When we applied it to our own nasal mucosa or palmar skin, it had a soothing non-irritant, lotion-like consistency.

Before testing, AM-301 and its matrix were brought to room temperature and vigorously agitated for 10 s. The amount of product tested on MucilAir inserts was calculated to roughly correspond to the amount that would be delivered to the nostril by a standard, commercial spray applicator (140 µL per actuation). Considering that the total adult nasal cavity surface area is 160 cm^2^ [57] and that a nasal spray coats the anterior third of the cavity [21], 5.3 μL of product should be tested per square centimeter of tissue. Since MucilAir inserts are 0.33 cm^2^, 10 μL of a 1:5 aqueous dilution of AM-301 or its matrix (diluted in water immediately before use) was tested per insert.

### 4.3. Safety Assay Design

AM-301 was tested for potential cytotoxic effects in a series of assays to determine the viability and function of the epithelial tissues used routinely for assessing the quality of MucilAir preparations. These assays, performed by Epithelix, included an assay for transepithelial electric resistance (TEER), which measures tight junction integrity [58]; an assay for lactate dehydrogenase (LDH) release into the basolateral medium, a standard measure of cytotoxicity; and an assay for cilia beating frequency (CBF), which is an index of the main function of airway cells, namely mucociliary clearance [59]. The three assays were conducted simultaneously on one set of 12 MucilAir inserts over a 4-day protocol with repeated apical application of the product, as described below and illustrated in Figure 4A (see “Safety assays” below for the individual methods).

Briefly, 3 days before use, the inserts were washed apically with 200 μL culture medium (10 min), and TEER was measured to verify their integrity. The apical medium was discarded, and the inserts were returned to the incubator for 3 days. On the first day of product testing (Day 0), the inserts were transferred to new 24-well plate with 500 μL/well fresh medium. The apical surface was treated in triplicate with 10 μL of a 1:5 aqueous dilution of AM-301 or its matrix or left untreated. The inserts were returned to the incubator for 24 h.

On Day 1, 200 μL of the medium was applied apically for TEER measurements. The apical medium was removed, and the inserts were transferred to a new 24-well plate with 500 μL/well fresh medium. The inserts were treated as on Day 0 and returned to the incubator. On Day 2, 200 μL of the medium was applied apically for TEER measurements. The apical medium was removed, and the basolateral medium was collected for LDH assays. The inserts were transferred to a new 24-well plate with 500 μL/well fresh medium, treated as on Day 0, and returned to the incubator. On Day 3, 200 μL of the medium was applied apically for TEER measurements. The apical medium was removed, and the inserts were transferred to new 24-well plate with 500 μL/well fresh medium; the inserts were treated as on Day 0 and returned to the incubator. On Day 4, cilia beating on the apical surface was observed and its frequency was quantified. Then, 200 μL medium was applied apically for TEER measurements. Finally, the basolateral medium was collected and assayed for LDH. 

#### 4.3.1. Transepithelial Electric Resistance (TEER) Assay

Transepithelial electric resistance (TEER) was measured with an EVOMX epithelial volt-ohm meter (World Precision Instruments). Briefly, 200 μL of the MucilAir culture medium (34 °C) was applied to the apical surface of each MucilAir insert. The electrodes were washed with 70% ethanol and then with medium prior to insertion into the apical and basolateral media. Resistance values (Ω) were measured at room temperature; they were corrected and converted to TEER (Ω cm^2^) using the following formula: TEER (Ω cm^2^) = (resistance (Ω) − 100(Ω)) × 0.33 (cm^2^)(1)
where 100 Ω is the resistance of the membrane and 0.33 cm^2^ is the surface area of the insert. The assay was performed in triplicate. Normal values for MucilAir are in the range 200–600 Ω cm^2^ [54].

#### 4.3.2. Lactate Dehydrogenase (LDH) Assay

Lactate dehydrogenase activity in the basolateral medium was assayed using the Cytotoxicity Detection Kit (cat. no. 4744934001, Roche Diagnostics GmbH, Mannheim, Germany,). For the high control, MucilAir inserts (*n* = 3) were treated apically with 100 μL of a lysis solution for 24 h in a tissue culture incubator, and the basolateral medium was collected. For the assay, 100 μL of the medium from each test sample and from the high controls was transferred to a 96-well plate; culture medium alone was used as the low control (*n* = 3). The reaction solution was added (100 μL/well), and the plate was incubated for 15 min in the dark at room temperature. Then, a stop solution was added (50 μL/well), and absorbance was read at 490 nm on a plate reader. Cytotoxicity was expressed as a normalized percentage relative to the high and low controls. Normal values for MucilAir inserts are ≤5%, which corresponds to a physiological turnover of cells in culture [54].

#### 4.3.3. Cilia Beating Frequency (CBF) Assay

Cilia beating frequency at the apical surface of MucilAir inserts was observed at room temperature under a Primovert inverted microscope (Zeiss, Jena, Germany) with a 10× objective. A Mako G-030B machine vision camera (Allied Vision Technologies, Stadtroda, Germany) mounted on the microscope was used to capture 256 movies (125 frames per second). The registrations were analyzed with Cilia-X software (Epithelix), which calculated the cilia beating frequency. Normal values of frequency for MucilAir are in the range of 4–8 Hz.

### 4.4. Efficacy Assays Design

AM-301 was subjected to a series of in vitro efficacy assays using MucilAir inserts at Texcell. The ability to prevent SARS-CoV-2 infection was tested in a prophylaxis assay, while the ability to reduce SARS-CoV-2 replication in already-infected tissues was tested after post-exposure treatment in a mitigation assay. For both assays, inserts were treated apically with 100 μL of a suspension of SARS-CoV-2 in a culture medium at a multiplicity of infection (MOI) of 0.5 for the WT strain and 0.1 for the Delta variant.

#### 4.4.1. Pre-Viral Load Protocol

In the pre-viral load assay (prophylaxis), MucilAir inserts were apically treated with test items for 10 min before exposure to SARS-CoV-2 (SARS-CoV2 WT: MOI = 0.5; SARS-CoV-2 Delta variant: MOI = 0.1), and viral replication was measured over 4 days with daily apical treatments with the saline solution, the matrix, or AM-301. In the experiment evaluating the ability of AM-301 to prevent SARS-CoV-2 WT infection (Figure 2A,C), TCID50 measurements were performed on three independent replicates per group. In the experiment evaluating the ability of AM-301 to prevent a SARS-CoV2 Delta variant infection, TCID50 measurements were performed on five independent replicates per group (Figure 2B,D).

Briefly, on Day 0, the inserts were transferred to a new 24-well plate with 500 μL/well fresh medium and washed apically by incubation with 200 μL medium for 20 min at 34 °C in a humidified 5% CO_2_ atmosphere; then, the apical medium was removed. The apical surface was then treated with 10 μL of a 1:5 aqueous dilution of AM-301 or of its matrix, or with 10 μL saline diluted 1:10 in water. After 10 min, without washing, the apical side was treated with 100 μL SARS-CoV-2 suspension (SARS-CoV-2 WT: MOI = 0.5; Delta variant: MOI = 0.1). Infection was allowed to proceed for 3 h in a 34 °C incubator and stopped by gentle washing of the apical surface with 200 μL medium (3 times). Viral replication was assessed immediately and daily over 4 days as follows:(1)*Viral sampling*: 300 μL of the medium was applied to the apical surface. After 20 min at 34 °C, the conditioned apical medium was collected and stored at −70 °C until analysis.(2)*Medium change*: Inserts were transferred to a new 24-well plate with 500 μL/well fresh medium. The conditioned basolateral medium was collected and stored at −70 °C for possible future analyses.(3)*Repeat treatment*: The apical surface was treated with a product or left untreated, as described above, and the inserts were returned to the 34 °C incubator.

Treatments were repeated daily, and sampling was performed only on Day 4.

#### 4.4.2. Post-Viral Load Protocol

In the post-viral load assay (mitigation), MucilAir inserts were apically infected with the SARS-CoV-2 WT or Delta variant, and viral replication was measured over 4 days with daily apical treatments of physiological saline, AM-301, or its matrix In the experiment evaluating the ability of AM-301 to mitigate SARS-CoV-2 WT infection (Figure 3A,C), TCID50 measurements were performed on three independent replicates per group. In the experiment evaluating the ability of AM-301 to prevent SARS-CoV2 Delta variant infection, TCID50 measurements were performed on five independent replicates per group (Figure 3B,D).

Briefly, on Day 0, the inserts were transferred to new 24-well plates with 500 μL/well fresh medium and washed apically by incubation with 200 μL medium for 20 min at 34 °C in a humidified 5% CO_2_ atmosphere; then, the apical medium was discarded. The apical surface was infected with 100 μL of SARS-CoV-2 suspension (SARS-CoV-2 WT: MOI = 0.5; Delta variant: MOI = 0.1); viral suspension was added to 12 inserts per protocol, while 3 inserts served as negative viral controls. Infection was allowed to proceed for 3 h in a 34 °C incubator and stopped by gentle washing of the apical surface with 200 μL medium (three times). Viral replication was assessed immediately by the addition of 300 μL of the medium to the apical surface and incubation for 20 min at 34 °C; the conditioned apical medium (Day 0) was collected and stored at −70 °C until analysis. The inserts were transferred to new 24-well plates with 500 μL/well of fresh medium and returned to the incubator. 

On Day 1, viral replication was assessed as on Day 0 for all inserts. All inserts were transferred to new 24-well plates with 500 μL/well of fresh medium. Inserts belonging to the mitigation arm of the study were treated with AM-301, its matrix, or physiological saline and returned to the incubator.

On Days 2 and 3, viral replication was assessed in all inserts; the conditioned medium (Days 2 and 3) was collected and stored at −70 °C. Inserts were transferred to new 24-well plates with fresh medium; treated with AM-301, its matrix, or saline; and returned to the incubator. On Day 4, viral replication was assessed in all inserts.

### 4.5. Viral Titer Assay

The viral titers were determined in the conditioned medium collected from the apical side of MucilAir inserts. The samples were prediluted 1:32 in DMEM containing 0.5 mg/mL gentamicin (to avoid any possible interference of the test products on cell growth). Then, using one 96-well plate per sample, serial three-fold dilutions were made in the same culture medium with eight replicates per dilution (10 dilutions per sample); eight wells received fresh medium (negative controls). A fixed volume (50 μL) from each well was transferred to sample titration plate, and 50 μL VERO cell suspension (10^5^ cells/mL in DMEM plus 4% FBS) was added per well. Plates were incubated at 37 °C in a 5% CO_2_ humidified atmosphere for 6 days. Then, a 0.2% crystal violet solution was added to stain DNA in live, adherent cells. The tissue culture infectious dose that killed 50% of cells (TCID_50_) was calculated using the Spearman–Kärber method.

### 4.6. Statistical Analyses

Data from the LDH and CBF analyses were compared using the unpaired *t*-test. The results from the TEER analyses were compared using two-way repeated measure ANOVA followed by post hoc Tukey’s test. The level of significance was set at *p* < 0.05. The results from the SARS-CoV-2 prophylaxis and mitigation experiments were analyzed using linear mixed-effect models with log-transformed data [35]. The minimal adequate mixed-effect model was achieved by top-down selection, adjusting for multiple comparisons. Analyses were performed using the lme4 package [34] as implemented in the programming language R [60].

## Figures and Tables

**Figure 1 ijms-23-04062-f001:**
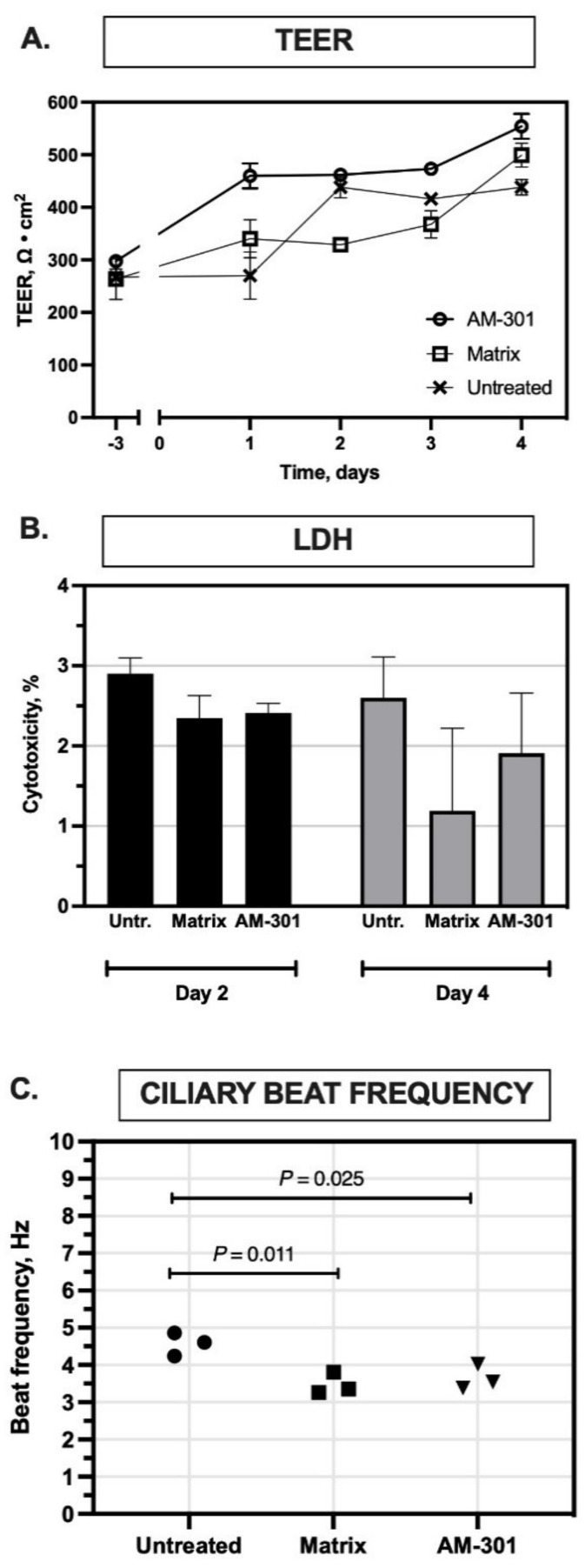
In vitro safety of AM-301 on MucilAir inserts (human nasal airway epitheliums). (**A**) Transepithelial resistance (TEER) over 4 days of exposure to AM-301 or its matrix lacking bentonite. (**B**) Lactate dehydrogenase (LDH) release assay for cytotoxicity. Data are expressed as a percentage of the amount of LDH released by lysed cells. (**C**) Ciliary beat frequency after 4 days of exposure to AM-301 or its matrix.

**Figure 2 ijms-23-04062-f002:**
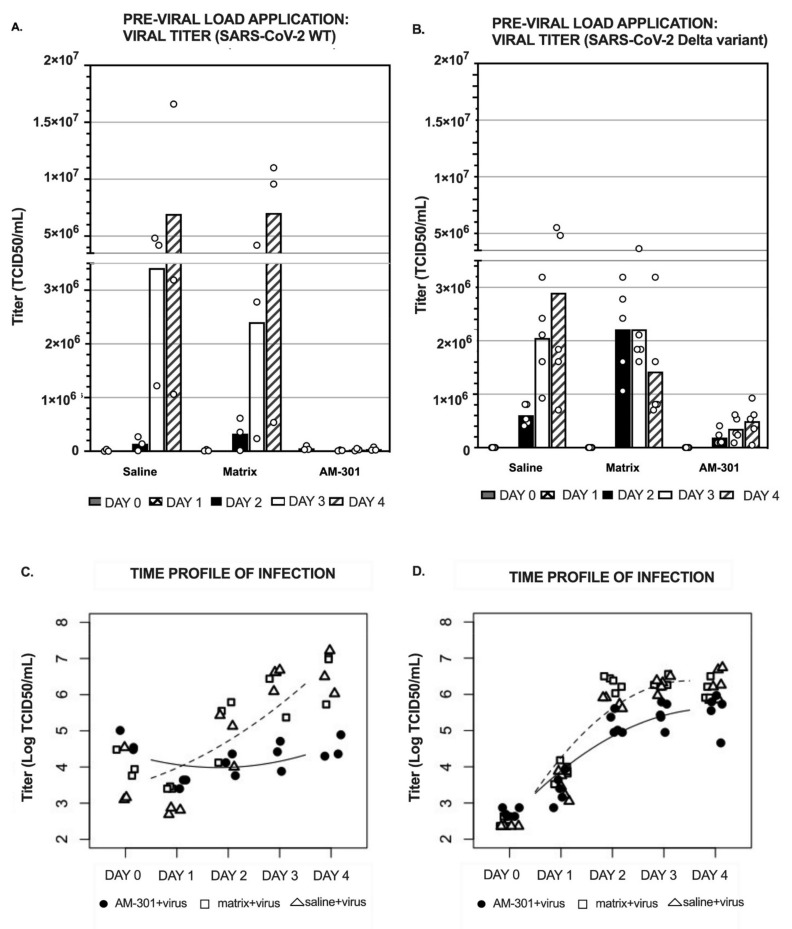
AM-301 as prophylaxis (pre-viral load application) against SARS-CoV-2 infection. MucilAir inserts were treated for 10 min with physiological saline, AM-301, or its matrix, followed by viral suspension for 3 h. Then, inserts were washed and incubated for up to 4 days with daily reapplication of saline, matrix, and AM-301. (**A**) Bar chart with mean values (bars) and individual data points (void circles) of inserts infected by SARS-CoV-2 WT (independent replicates, *n* = 3 per group). (**B**) Bar chart with mean values (bars) and individual data points (void circles) of inserts infected with SARS-CoV-2 Delta variant (independent replicates, *n* = 5 per group). (**C**,**D**) Linear mixed-effects model. The log-linear scatter plot shows individual log-transformed data and regression lines for negative control samples (saline- and matrix-treated inserts; dashed line) and for AM-301-treated inserts (continuous line). Black bullets represent the AM-301+virus group, the white squares represent the matrix+virus group, and the white triangles represent the saline + virus group. The viral growth kinetics in saline and matrix-treated groups (controls) were not significantly different from each other.

**Figure 3 ijms-23-04062-f003:**
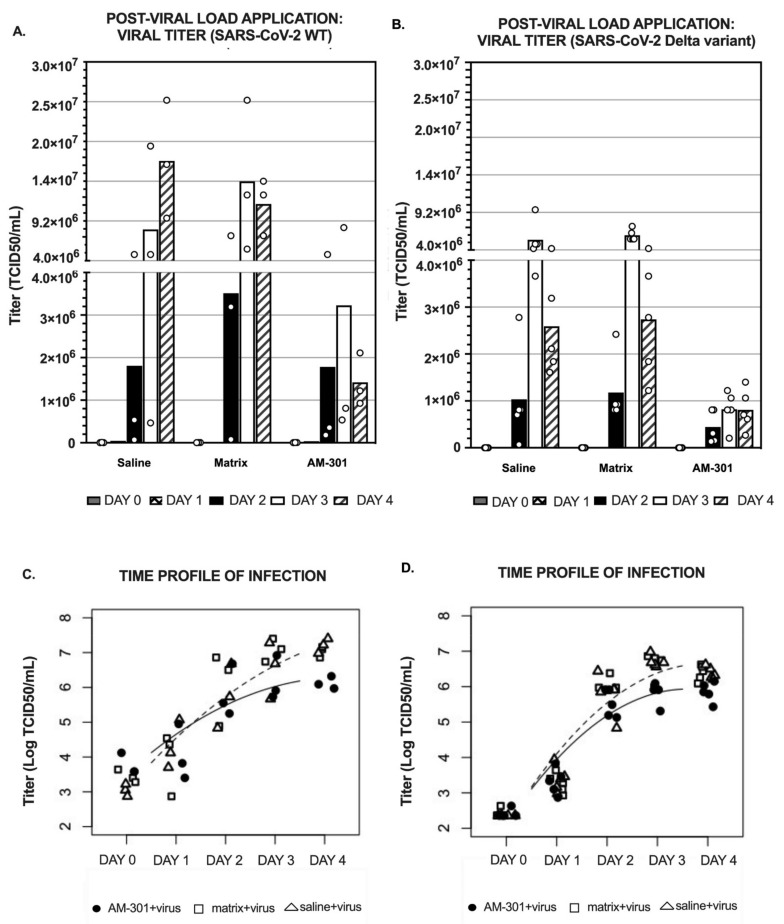
AM-301 as mitigation (post-viral load application) of SARS-CoV-2 infection. Test substances were applied 24 h after the start of the experiment. (**A**) Bar charts with mean values and individual data points of inserts infected by SARS-CoV-2 WT (independent replicates, *n* = 3 per group). (**B**) Bar chart with mean values and individual data points of inserts infected by SARS-CoV-2 Delta variant (independent replicates, *n* = 5 per group) (**C**,**D**) Linear mixed-effects model. The log-linear scatter plot shows individual log-transformed data and concave curves for negative control samples (saline- and matrix-treated inserts; dashed curve) and for AM-301-treated inserts (continuous curve). The model shows a deceleration in exponential growth, as is typically observed for sigmoidal behaviors.

**Figure 4 ijms-23-04062-f004:**
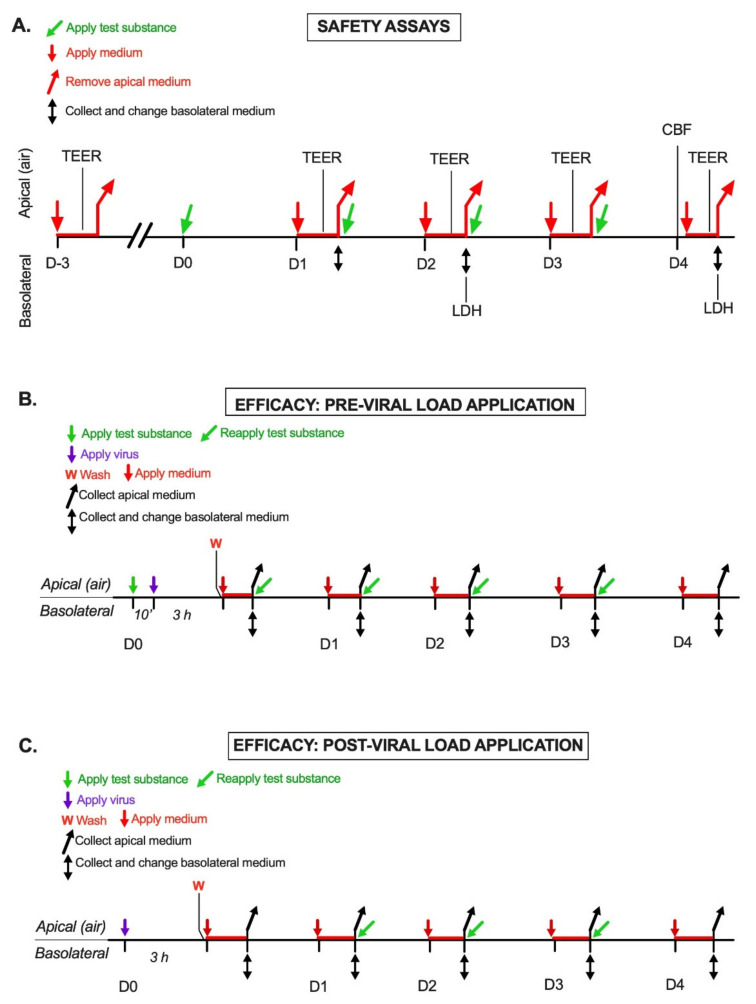
Schematics of experimental protocols. (**A**) Safety assay experimental design. CBF, cilia beating frequency; LDH, lactate dehydrogenase; TEER, transepithelial electric resistance. (**B**) Efficacy assay: Pre-viral load application (Prophylaxis). (**C**) Efficacy assay: Post-viral load application (Mitigation).

## Data Availability

Data are included in the article.

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
