# Peer review of "Drug-Free Nasal Spray as a Barrier against SARS-CoV-2 and Its Delta Variant: In Vitro Study of Safety and Efficacy in Human Nasal Airway Epithelia"

_ijms, 2022, doi:10.3390/ijms23074062_

Round 1
Reviewer 1 Report
See file

Reviewer 2 Report
The authors investigated the safety and efficacy of AM-301 (bentonite-containing nasal spray) against SARS-Cov-2 infection in vitro models and found that AM-301 did not have cytotoxicity to nasal epithelium while reduced the virus replication by 99%. The in vitro experiments were well designed, and the results were presented and interpreted appropriately.
COVID-19 is a disease caused by overreacted inflammation response in host lung and other organs upon virus invasion and stimulation. Bentonite was reported of pro-inflammatory properties in rat lungs (PMID: 31991556). Therefore it is critical to investigate whether the AM-301 has any impact to the inflammatory response (such as inflammatory cytokine levels) in the testing models. The biodegradability and decomposition of bentonite in the testing models also needed to be investigated as repeat application of bentonite could be problematic to be cleared from the nasal-bronchus-lung system. However it might be beyond the scope of current study.
Reviewer 3 Report
Fais and colleagues undertook an interesting study assessing the safety and efficacy of a newly developed nasal spray against SARS-CoV-2 in an in vitro model of primary human nasal airway epithelium. The novelty of this study cannot be overlooked. As emphasized by the authors, in addition to current non-pharmaceutical interventions, a safe, effective and potentially cost-effective and importantly widely accessible intervention is needed.
Harnessing the biophysical properties of Bentonite as an important feature of this intervention must be applauded. This novel yet simplistic tool will undoubtedly pave the way for further research and possibly improvements in said intervention. Acknowledging the nasal airway as an important site of intervention is also a strength of their hypothesis.
This study is therefore highly valuable however listed below are some suggestions that warrant further clarification and improvement to highlight and most importantly solidify the arguments presented in the paper.
Minor comments
- The study assessed the safety and efficacy in well thought of experiments that seem appropriate for the study. Importantly a strength of the study was that this was built on previous safety data by the group, from individuals with allergic rhinitis
- Based also on the data they have presented, appropriate statistical tests have been performed to ascertain any differences between the different interventions/groups.
- It is my understanding at line 200 the authors suggest that despite applying the spray several hours after exposure to virus it was highly effective. As such it is advised they reword this sentence and replace ‘applied only many hours after’ with perhaps ‘applied several hours post-exposure’. Simple grammar fix will really sell their message here that this intervention is also highly effective at preventing virus egress.
Major comments
- Findings of the efficacy of mitigation of viral egress are convincing however it would benefit the authors to include a few more repeats particularly Figure 1C. It is unclear in the legend or the text how many repeats were performed for this experiment. Do the 3 symbols represent 3 independent repeats or were these experiments conducted at the same time, with the same cell passage (on the cell line)? It is recommended that the authors clarify this or add additional repeats if the work is from just one experiment. It is also unclear if the data represent mean or median. It would suffice to add this in the figure legend.
- Appropriate y-axis scales should be applied to Figure 2A&B and Figure 3A&B to ensure that points at or near the 0-axis are also visible. Perhaps include in the scale values below 0 to highlight this.
- Clearly viral load for Delta is significantly higher than WT however it might benefit also displaying Delta data as they have done WT data to allow uniformity with the presentation. While the data is self-explanatory and convincing, at present does not allow fair comparisons to be made.
The very important conclusion the authors state that this intervention not only has implications for SARS-CoV-2 but also other airborne viruses highlights the importance of this work. However as stated above, I would recommend some minor revisions, particularly with regards to additional experiments for Figure 1C to finalise this paper.
Round 2
Reviewer 1 Report
This manuscript is still a commercial for their product.
Reject is my suggestion.Author Response
Reply not to be filled in as per Assistant Editor's (Milica Spasojevic) recommendation.
Reviewer 3 Report
Thank you kindly for addressing the comments/suggestions I made.